# Simultaneous Natural Deep Eutectic Solvent-Based Ultrasonic-Assisted Extraction of Bioactive Compounds of Cinnamon Bark and Sappan Wood as a Dipeptidyl Peptidase IV Inhibitor

**DOI:** 10.3390/molecules25173832

**Published:** 2020-08-23

**Authors:** Islamudin Ahmad, Ayun Erwina Arifianti, Aditya Sindu Sakti, Fadlina Chany Saputri, Abdul Mun’im

**Affiliations:** 1Department of Pharmaceutical Sciences, Faculty of Pharmacy, Universitas Mulawarman, Samarinda, East Kalimantan 75119, Indonesia; islamudinahmad@farmasi.unmul.ac.id; 2Department of Pharmaceutical Technology, Faculty of Pharmacy, Universitas Indonesia, Depok, West Java 16424, Indonesia; ayun.arifianti@farmasi.ui.ac.id; 3Graduate Program of Herbal Medicine, Faculty of Pharmacy, Universitas Indonesia, Depok, West Java 16424, Indonesia; munim@farmasi.ui.ac.id (A.S.S.); fadlina.chany@farmasi.ui.ac.id (F.C.S.); 4Department of Pharmacognosy-Phytochemistry, Faculty of Pharmacy, Universitas Indonesia, Depok, West Java 16424, Indonesia

**Keywords:** Cinnamon bark, dipeptidyl peptidase IV, natural deep eutectic solvent, sappan wood, ultrasonic-assisted extraction

## Abstract

Cinnamon bark (*Cinnamomum burmannii*) and sappan wood (*Caesalpinia sappan*) have been reported to be beneficial for Type-2 Diabetes Mellitus (T2DM) and the combination is commonly used by Indonesian herbal industries. In the present study, the simultaneous extraction of bioactive compounds from both plants was conducted using natural deep eutectic solvent (NADES), their content analyzed using high-performance liquid chromatography (HPLC), and their dipeptidyl peptidase IV (DPP IV) inhibitory activity evaluated. An additional in silico molecular docking analysis was conducted to ensure their activity. The results showed that NADES (with a composition of choline chloride–glycerol) extraction from cinnamon and sappan wood had DPP IV inhibitory activity of 205.0 and 1254.0 µg/mL, respectively. Brazilin as a marker substance from sappan wood was responsible for the DPP IV inhibitory activity, while none of the marker substances chosen for cinnamon bark (*trans*-cinnamaldehyde, coumarin, and *trans*-cinnamic acid) were found to have significant DPP IV inhibitory activity. These results were confirmed by molecular docking conducted in brazilin, *trans*-cinnamaldehyde, coumarin, and *trans*-cinnamic acid.

## 1. Introduction

Diabetes, as a metabolic disease, is indicated by hyperglycemia resulting from defects in insulin secretion, insulin action, or both. Type 2 Diabetes Mellitus (T2DM), as the most prevalent type, is caused by both an insulin resistance and an inadequate compensatory insulin secretory response [1]. Sulfonylureas, biguanides, meglitinide, thiazolidinedione (TZD), α-glucosidase inhibitors, sodium–glucose cotransporter (SGLT2) inhibitors, and dipeptidyl peptidase IV (DPP IV) inhibitors are the dominant oral antidiabetic medications [2]. DPP IV inhibitors are superior for elderly and cardiac disease patients compared with other drugs due to the low incidence of hypoglycemia [3].

DPP IV inhibitors include sitagliptin, saxagliptin, vildagliptin, linagliptin, and alogliptin. The gliptins have in clinical trials, unfortunately, caused common adverse reactions, such as headache, nasopharyngitis, and upper respiratory tract infection [2]. The development of DPP IV inhibitors from natural products is needed to obtain safer drugs. Several plants have been reported to be beneficial for T2DM and are commonly used by Indonesian herbal industries. Cinnamon (*Cinnamomum burmannii*) bark and sappan (*Caesalpinia sappan*) wood have been reported to have insulin-induced glucose uptake enhancement in the sequence 98.0 ± 15.8 and 161.1 ± 19.9 mg/mL [4].

Some studies have revealed that sappan and cinnamon have beneficial effects on diabetes mellitus [5,6]. Sappan wood works through the mechanism of α-glucosidase and DPP IV inhibition, as well as inhibition of gluconeogenesis and hepatic glycolysis [5,7,8]; whereas cinnamon works through the mechanism of activating glucose transporter type-4 (GLUT-4) and α-glucosidase inhibition [9,10]. However, in previous studies the extraction of sappan wood with green solvent using ionic liquid only produced a small amount of yield (0.9%) [8]. In addition, the mechanism of action of anti-diabetic cinnamon through the inhibition pathway DPP IV has not been disclosed.

To obtain beneficial effects from natural products, an extraction strategy is needed [11]. Based on conventional solvents comparisons, the use of a natural deep eutectic solvent (NADES) has some advantages: it is biodegradable, non-toxic, non-volatile, not frozen at low temperatures, has high extraction ability, a relatively low price, and simple preparation [12,13]. Besides that, the use of the ultrasonic-assisted extraction (UAE) method is one of the non-conventional extraction methods that can extract target secondary metabolites with high reproducibility, shorter extraction times, and minimal solvent use [14]. Some studies have reported the successful use of the UAE method in extracting secondary metabolites from plants, including the extraction of phenols from sunflower [15] and black locust (*Robiniae presudoacaciae*) [16], the extraction of polyphenols from wheatgrass (*Triticum aestivum* L.) [17], the extraction of flavonoids from bamboo leaves [18], and caffeine and chlorogenic acid extraction from coffee beans (*Coffea arabica* L.) [19].

In previous studies, optimum conditions were obtained in the extraction process of marker compounds from cinnamon bark and sappan wood [20]. However, research has not been reported related to simultaneous NADES-based UAE (NADES-UAE) of bioactive compounds from the combination of cinnamon bark and sappan wood as DPP IV inhibitors. Therefore, this study aims to determine the effect of using the NADES-UAE method simultaneously on the biomarker compounds and their activity as DPP IV inhibitors in both single and combined forms.

## 2. Results and Discussions

### 2.1. Simultaneous NADES-UAE Process and Marker Compound Determination

In this study, the use of NADES -UAE was successful in extracting marker compounds contained in the combination of cinnamon bark and sappan woods (1:1 *w*/*w*). NADES-based extracts contain higher marker compounds compared with the conventional reflux method. In the cinnamon bark were obtained *trans*-cinnamaldehyde and coumarin contents of 108.5 and 273.7 µg/mL (reflux) and 877.8 and 968.8 µg/mL (NADES-UAE), respectively, whereas in sappan woods was obtained brazilin content of 62.3 µg/mL (reflux) and 628.4 µg/mL (NADES-UAE).

According to the resulting study (previously reported by Sakti et al. 2019), the optimum extraction method for cinnamon bark and sappan wood was 20% *v*/*v* water content in NADES, a sample–NADES ratio of 1:4 *w*/*w*, and a choline chloride–glycerol ratio of 2:1 *w*/*w* (Method I) and 47.6% *v*/*v* water content in NADES, a sample–NADES ratio of 2:1 *w*/*w*, and a choline chloride–glycerol ratio of 1:2 *w*/*w* (Method II), respectively [20]. Furthermore, both methods were applied for the simultaneous extraction of a combination of cinnamon bark and sappan woods at a ratio of 1: 1 *w*/*w*. The obtained extract was determined by the marker compound content, as presented in Figure 1. NADES (with a choline chloride–glycerol composition), compared with using water as a solvent, was able to extract the content of marker compounds in these plant combinations (people use water as a solvent for daily use traditionally).

The results of the determination of the marker compound content showed that NADES-UAE was better in extracting the marker compound from the sample combination compared with H_2_O-UAE. There was a significant difference between the marker compound content in the NADES-UAE extract and that in the H_2_O-UAE extract. In both methods, the *trans*-cinnamaldehyde and coumarin contents in NADES extracts are three times greater than those in water extracts; while the brazilin content in Method I is nine times greater than that in the water extract and in Method II six times greater. The data show that the choline chloride–glycerol-based NADES used has the potential to increase the content of marker compounds extracted from the combination of cinnamon bark and sappan wood.

The determination of compound content was performed simultaneously using high-performance liquid chromatography (HPLC). Figure 2 shows that the used gradient system successfully separates the marker compounds contained in a combination of sappan wood and cinnamon bark. Based on Figure 2, target marker compounds were detected in one chromatogram simultaneously including brazilin (Rt = 7.665 min), coumarin (Rt = 16.786 min), and *trans*-cinnamaldehyde (Rt = 20.237 min). In the determination of simultaneous contents, a combination was used of aqueous phases containing acetic acid of 0.04% to 0.3% and acetonitrile. The used content determination method has excellent precision and reproducibility. The precision of the determination method can be observed from the relative standard deviation (coefficient of variation) generated by repeated sample testing.

### 2.2. In Vitro DPP IV Inhibitory Activity Assay

DPP IV or CD26 (cluster of differentiation 26) is a specific proteolytic enzyme that cleaves proline and alanine amino acid residues. One of its physiological functions is to promote blood glucose homeostasis by degrading incretin hormones such as glucagon-like-peptide-1 (GLP-1) and glucose-dependent insulinotropic polypeptide (GIP). In the case of type-2 diabetes (uncontrolled elevation in postprandial glucose), those two hormones have a longer onset through the inhibitor mechanism of DPP IV. Cayman’s DPP IV Inhibitor Screening *Assay* provided a convenient fluorescence-based method for screening (DPP-IV) by using the fluorogenic substrate Gly-Pro-Aminomethylcoumarin (AMC). This assay was able to measure fluorescence from the free AMC group which was released through the cleavage of peptide bond by DPP IV. The assay was analyzed using an excitation wavelength of 350–360 nm and an emission wavelength of 450–465 nm [21].

The test results of DPP IV inhibitory activity showed that all samples had a percent inhibition above 50% (Figure 3); a retest was conducted in some different concentrations in order to obtain an IC_50_ value (Table 1). The data showed that brazilin as a marker substance from sappan wood was responsible for the DPP IV inhibition. Interestingly, all marker substances chosen for cinnamon barks (*trans*-cinnamaldehyde, coumarin, and *trans*-cinnamic acid) were found to have no DPP IV inhibitory activity. This result was confirmed by molecular docking conducted in brazilin, *trans*-cinnamaldehyde, coumarin, and *trans*-cinnamic acid. However, the DPP IV inhibitory activity of the NADES extract of cinnamon bark was more potent than that of the NADES extract of sappan wood. The IC_50_ value showed that the NADES extract of cinnamon bark had 205.0 µg/mL, while the NADES extract of sappan wood had 1254.0 µg/mL. Further investigation was needed to reveal the marker substance in cinnamon responsible for DPP IV inhibitory activity.

The combined NADES extract of cinnamon bark and sappan wood in Method 1 (the extraction method for maximizing brazilin) showed more potent DPP IV inhibitory activity than the combined NADES extract of cinnamon bark and sappan wood in Method 2 (the extraction method for maximizing *trans*-cinnamaldehyde and minimize coumarin), Table 1. Compared with a conventional extract, the ethanolic reflux extract from sappan wood had a higher IC_50_ value than the NADES extract. This result was due to the unavailability of a yield value determination from the NADES extract since it was unable to evaporate. The calculation of the NADES extract’s IC_50_ value was done through a concentration percentage. Further investigation was needed for separation in the NADES extraction in order to obtain an accurate yield value and concentration of extract. Thus, the yield value and the concentration of the extract would be able to more accurately confirm the DPP IV inhibition potential in the NADES extract. On the other hand, a IC_50_ value comparison could not be made in the cinnamon bark extract since the ethanolic reflux extract only showed a DPP IV inhibitory activity of 19.7% (for 500 µg/mL).

### 2.3. In Silico Molecular Docking Study

Based on the incredible in vitro activity against DPP IV, a molecular docking study was conducted to support the potency of its inhibition. The inhibition constant (CI) is a reverse equivalent to the affinity of marker compounds against the enzyme. Therefore, marker compounds that had high DPP IV inhibition activity would have a lower value of the inhibition constant. Brazilin was the marker that had the highest affinity against DPP IV, so it was predicted to have the highest inhibitory activity among all markers. This was based on the re-docking result of the native ligand demonstrating the root square mean deviation (RSMD) value from the lower free binding energy ΔG of backbone atoms and their interaction on the active site of the receptor. The RSMD value obtained was 0.448 Å (<2 Å) with a ΔG of -9.60 kcal/mol on 43 clusters run 100 times (Table 2), which indicates that the docking results are valid. 

In line with the CI, the lower free binding energy or Gibbs energy (ΔG) would result in the more stable interaction between ligand and macromolecule receptor. Two-dimensional (2D) visualization of marker interaction can be seen in Figure 4. Marker compounds in cinnamon bark had lower total hydrogen bonds on a DPP IV active site compared with brazilin as a marker in sappan wood. Table 2 showed that the brazilin with ΔG values of −6.35 (with an inhibition constant of 22.06 µM) was lower than other marker compounds. Besides, brazilin has the same interaction on the active site of the macromolecule receptor as the native ligand interaction. These results were consistent with the in *vitro assay* result.

In protease enzymes of macromolecule receptors, such as DPP IV, the active sites are known as subsites that are binding sites for the peptide substrate. The amino acid residue in the peptide substrate was given a number based on the peptide cleavage point by the macromolecule enzyme. In DPP IV, these were two cleavage points P1, P1′ and P2, P2′, while the given name in the subsite was S1, S1′ and S2, S2′. In DPP IV, the *N*-terminal from the peptide substrate will be familiar to Glu205, Glu206, and Ser630 so that the cleavage will happen in the penultimate position of the *N*-terminus from the peptide (P1). Besides *subsite* S1 and S2, *subsite* S2 *extensive* (S2 ext.) included Val207, Ser209, Phe357, and Arg358. Several inhibitors that can be bonded in S2 ext. resulted in a stronger inhibitory effect, but under normal conditions subsite S2 ext. cannot be bonded with the substrate. Strong hydrophobic interactions at the S2 subsite are more crucial than the interactions at the S2 ext. subsite for an efficient DPP IV inhibitor drug [22].

A standard DPP IV inhibitor that works through the class III inhibitory mechanism is the piperazine group, such as sitagliptin and teneligliptin. The piperazine substituent in that inhibitor works as an anchor lock domain in subsite S2 ext. In brazilin, the hydroxyl group in C1 and C8 as the anchor lock domain binds with amino Arg358 and Ser209 from subsite S2 ext. Marker substances from cinnamon bark, such as coumarin, *trans*-cinnamaldehyde, and *trans*-cinnamic acid, did not show significant interaction with the subsite DPP IV. Even *trans*-cinnamic acid did not interact with subsite DPP IV; therefore, *trans*-cinnamic acid did not compete with the substrate to be in subsite DPP IV. Further investigation is needed to investigate whether the hydrogen bond between *trans*-cinnamic and amino acid Glu527, Lys554, and Arg560 in DPP IV impacted upon DPP IV activity.

*Trans*-cinnamaldehyde did not have a hydrogen bond with subsite DPP IV, but there was some non-hydrogen interaction with subsites S1, S2, and S2′. Each group in an α-pyrone ring from coumarin had a hydrogen bond with Tyr631 from subsite S1′ DPP IV. Non-hydrogen interaction also happened in the amino acid from subsites S1, S1′, and S2′. Overall, the result from molecular docking showed that brazilin, as the marker of sappan wood, was predicted to have DPP IV inhibitory activity. In contrast, other marker substances did not have significant DPP IV inhibitory activity.

## 3. Materials and Methods

### 3.1. Materials

The material samples used in this study include sappan wood (*C. sappan* L.) obtained from Magelang in the Central Region of Java, and cinnamon bark (*C. burmanii* Blume) from Padang, West Sumatera Indonesia. Both plants were authenticated at the Herbarium Bogoriense, Botanical Gardens, Bogor, West Java, Indonesia. Choline chloride was purchased from Xi’am Rongsheng, Hangzhou, China; glycerol was purchased from PT Molex Ayus, Tangerang, Indonesia; demineralized water and 96% *v*/*v* ethanol were purchased from Brataco, Bogor, Indonesia; methanol (HPLC grade), acetonitrile (HPLC grade), glacial acetic acid, brazilin, coumarin, and *trans*-cinnamaldehyde standard were purchased from Sigma Aldrich through PT. Elo Karsa Pratama, South Jakarta, Indonesia; double-distilled water (ddH_2_O) was purchased from PT. Ikapharmmindo Putramas, East Jakarta, Indonesia; and DPP IV kits were purchased from Cayman Chemical, Ann Arbor, USA.

### 3.2. Equipment

High-performance liquid chromatography (Shimadzu LC-20AT, Japan), a microplate reader (Glomax Promega, Madison, WI, USA), micropipettes (10–100 µL and 100–1000 µL) (Socorex, Switzerland), micropore filter paper (0.45 µm) (Whatman, Marlborough, MA, USA), a rotary vacuum evaporator (Rotavapor^®^ R-215, Buchi, Flawil, Switzerland), an ultrasonic bath (Krisbow, Jakarta, Indonesia), digital scales (Vibra HT, Tokyo, Japan), a C18 column (4.6 mm × 150 mm with a pore size of 10 nm) (Inertsil, Fukushima, Japan), a microsyringe (Hamilton, Reno, NV, USA), a centrifuge (Hettich Zentrifugen, Tuttlingen, Germany), a syringe filter (Agilent, Santa Clara, CA, USA), a magnetic stirrer (IKA^®^ C-MAG HS 7, Wilmington, NC, USA), ChemOffice Pro v15.00 PerkinElmer, Discovery Studio Visualizer, OpenBabel GUI, Phyton Molecular Viewer (PMV 1.5.6), and Autodock v4.2.6 and AutodockTools (http://autodock.scrips.edu/).

### 3.3. Conventional and Non-Conventional Extraction Process

Reflux extraction was conducted using a conventional method to obtain comparative data for NADES-based extraction. The reflux extraction method used 96% *v*/*v* ethanol (with a solid-to-liquid ratio of 10 mL/g) at 80 °C for three cycles. The obtained filtrates were combined and then evaporated using a rotary vacuum evaporator at 40 °C. The concentrated filtrates were collected and stored in a refrigerator at 20 °C for use in further analysis.

NADES was prepared by mixing choline chloride with glycerol in a beaker glass covered with parafilm using a magnetic stirrer at 80 °C with the speed of 900 rpm until the mixture turned into a clear liquid solution [19]. The NADES extraction process was conducted by an ultrasonicator with a power of 35 W and frequency of 42.000 Hz for 30 and 50 min for cinnamon bark and sappan wood, respectively [20,23,24]. The powder samples were mixed with demineralized water and NADES. The mixtures were then centrifuged for 10 min at 3.283 g to separate the NADES liquid extract from the waste. The liquid solution was filtered using 0.45 μm Whatman micropore filter paper. The filtrate was collected and stored in a refrigerator at 20 °C until further analysis.

### 3.4. Simultaneous NADES-UAE Process for Extraction of Marker Compounds from a Combination of Samples

The simultaneous NADES-UAE process was performed according to the previous study [20]; two optimum extraction methods were obtained to extract marker compounds contained in sappan and cinnamon using NADES with a choline chloride–glycerol composition. Both methods were used to extract the target marker compound from a combination of sappan wood and cinnamon bark (1:1 *w*/*w*) as presented in Table 3.

### 3.5. Analysis of Marker Compounds

Analysis of marker compounds was performed using high-performance liquid chromatography (HPLC) based on a previous study [20], with 280 nm UV-Vis detection. Briefly, a total of 100 µL of samples was dissolved in 80% *v*/*v* ethanol. The solution was homogenized with a shaker and filtered using a 0.45 µm syringe filter. The sample solution was stored in a tightly closed vial protected from light. A 20 µL sample solution was injected. The mobile phase was prepared including ddH_2_O (containing 0.3% acetic acid) and acetonitrile (85.5:14.5 *v*/*v*) for sappan wood and ddH_2_0 (containing 0.04% acetic acid) and acetonitrile (40:60 *v*/*v*) for cinnamon bark with flow rate of 1.0 mL/min (an isocratic system). Moreover, the mobile phase for the simultaneous analysis of the cinnamon bark and sappan wood extract combination was prepared using an elution gradient consisting of water containing acetic acid (A) and acetonitrile (B), as can be seen in Table 4.

### 3.6. In Vitro Dipeptidyl Peptidase IV Activity Assay

The Cayman DPP IV Inhibitor Screening *Assay* was used to measure the DPP IV inhibitor activity. The fluorogenic substrate Gly-Pro-Aminomethylcoumarin (AMC) was determined using GloMax^®^ Discover GM 300 with an excitation wavelength of 360 nm and an emission wavelength of 460 [21,22,25]. All reagents were prepared based on the protocol from Cayman. The extracted sample (10 µL) was mixed with 30 µL of buffer solution and 10 µL of the DPP IV enzyme. Sitagliptin was used as a positive control inhibitor, and the enzyme solution without sample was used as a negative control inhibitor. Subsequently, 50 μL of AMC substrate was added. The reaction was initiated after adding 50 µL of substrate to a 96-well microplate and incubated at 37 °C for 30 min. Each test sample was analyzed in triplicate. Percent inhibition was calculated using the following formula:(1)$$\%\;\mathrm{Relative}\;\mathrm{Inhibition}=\left(\frac{\mathrm{enzyme}\;\mathrm{activity}-\mathrm{inhibitor}\;\mathrm{activity}}{\mathrm{enzyme}\;\mathrm{activity}}\right)\;\times \;100$$

Moreover, the IC_50_ was calculated using linear regression based on some different concentrations of each sample.

### 3.7. In Silico Molecular Docking Analysis

In order to support the potency of its inhibition against DPP IV, an in silico molecular docking analysis was performed using Autodock 4.2.6 based on its protocols [26,27]. The ligand structure of markers was downloaded from https://zinc.docking.org, which were brazilin (ZINC00899553); *trans*-cinnamaldehyde (ZINC13523661); coumarin (ZINC00074709); and *trans*-cinnamic acid (ZINC16051516). The X-ray structure of DPP IV (PDB ID: 1X70) was downloaded from the Protein Data Bank (https://www.rcsb.org). A native ligand and a protein receptor were separated using Phyton Molecular Viewer (PMV 1.5.6). Water molecules were eliminated and protonated from the macromolecule complex; then, Gasteiger charges were added to each ligand atom. The native ligand was re-docked to obtain the best docking on the protein’s binding site using a Lamarckian Genetic Algorithm (LGA) based on the lowest free energy of binding or Gibbs energy (ΔG). The auto grid program was applied to determine the position of a grid. The central grid of the native ligand was placed using a box size of 52 Å × 28 Å × 26 Å and a grid center of 40.926 Å × 50.522 Å × 35.031 Å with a spacing of 0.375 Å. The in silico molecular docking simulation was performed using the Autodock 4.2.6 program with 100 runs; the docking results were visualized using Accelrys Discovery Studio Visualizer 4.0 [28].

## 4. Conclusions

Choline chloride–glycerol-based NADES extraction from cinnamon bark and sappan wood had a DPP IV inhibitory activity of 205.0 µg/mL and 1254.0 µg/mL, respectively. Brazilin as a marker substance from sappan wood was responsible for the DPP IV inhibition, but all marker substances chosen for cinnamon bark (*trans*-cinnamaldehyde, coumarin, and *trans*-cinnamic acid) were found to have no significant DPP IV inhibitory activity. This result was confirmed by the in silico molecular docking conducted in brazilin, *trans*-cinnamaldehyde, coumarin, and *trans*-cinnamic acid. In addition, it seems interesting to further study the cinnamon bark extract to identify the active DPP IV inhibiting component.

## Figures and Tables

**Figure 1 molecules-25-03832-f001:**
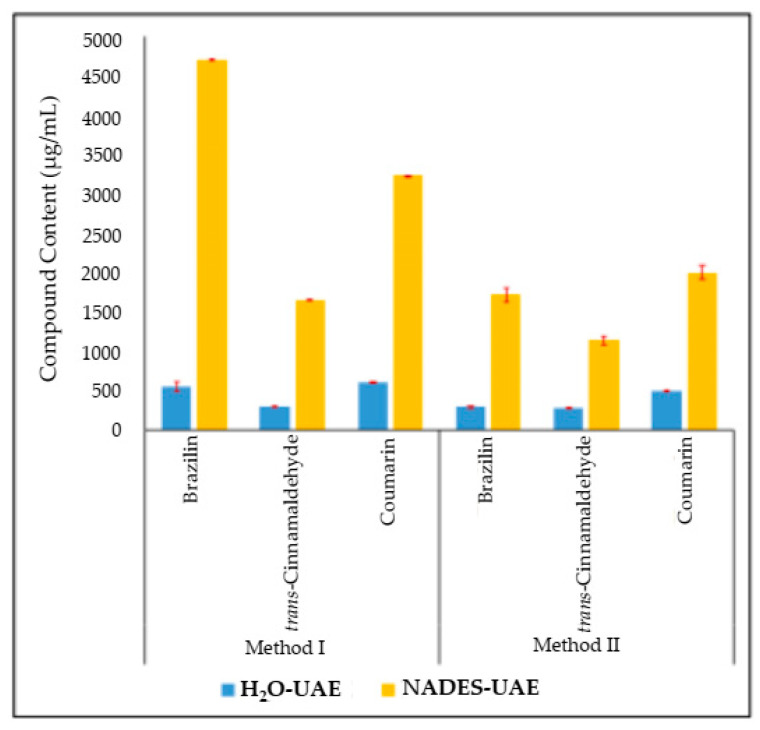
The ratio of compound content from extract solution using conventional solvent (H_2_O) and natural deep eutectic solvent (NADES).

**Figure 2 molecules-25-03832-f002:**
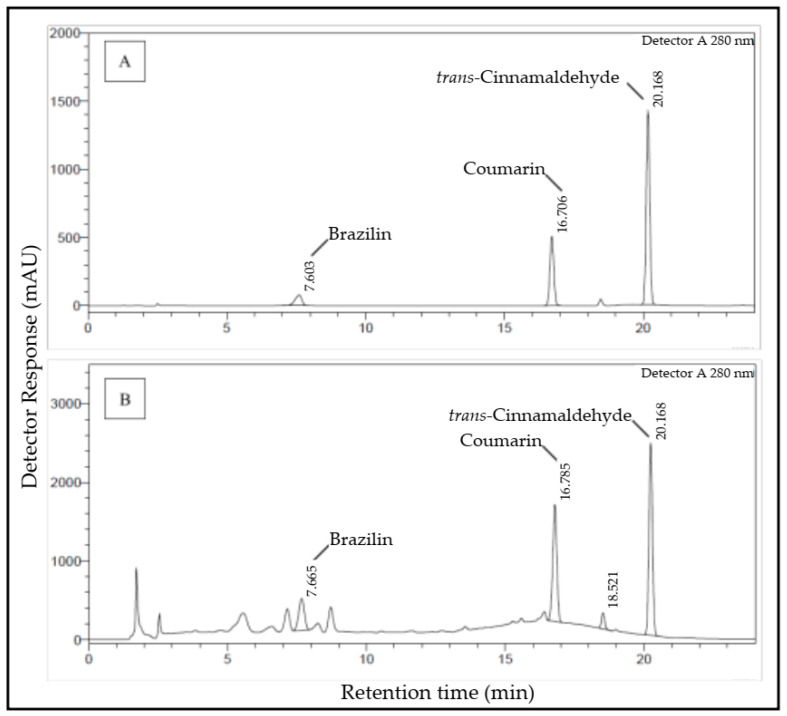
Chromatogram shows (**A**) a 100 µg/mL mixture of brazilin, coumarin, and *trans*-cinnamaldehyde standard, and (**B**) an NADES extract combination of cinnamom bark and sappan wood.

**Figure 3 molecules-25-03832-f003:**
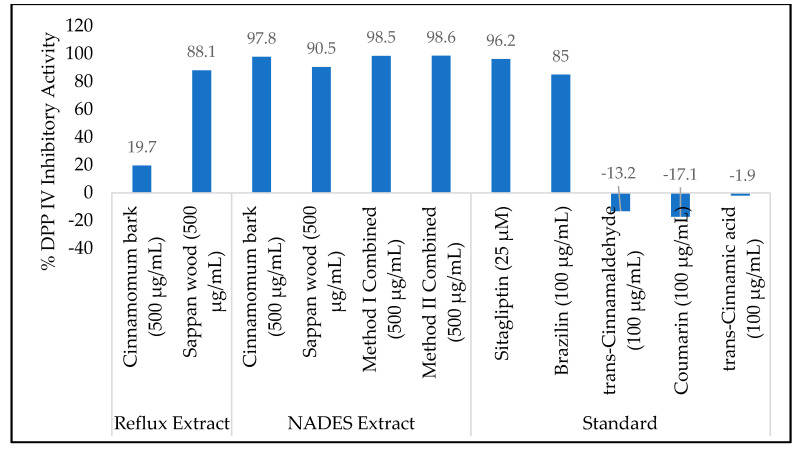
Dipeptidyl peptidase IV (DPP IV) Inhibitory Activity of extracts and standards.

**Figure 4 molecules-25-03832-f004:**
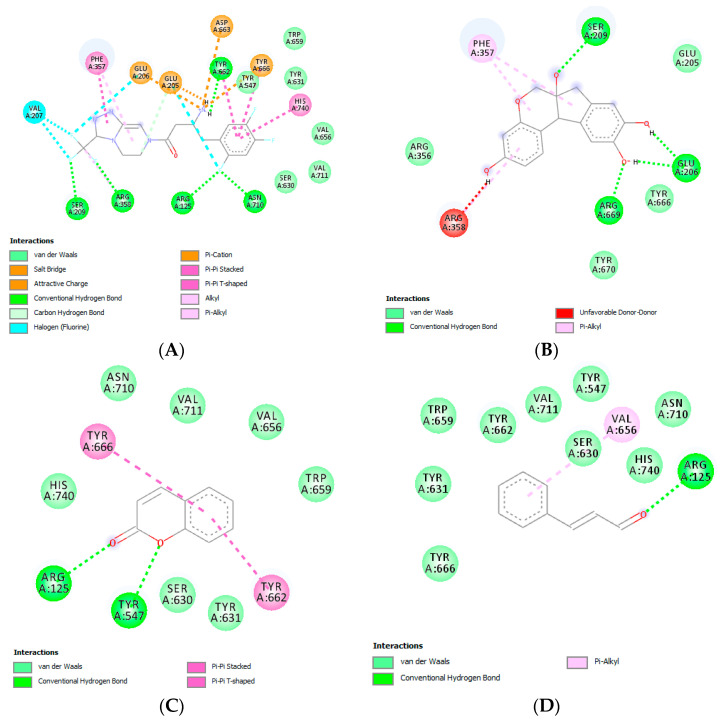
Ligand–Receptor Interaction: (**A**) Sitagliptin; (**B**) Brazilin; (**C**) Coumarin; (**D**) *trans*-cinnamaldehyde; and (**E**) *trans*-cinnamic acid.

**Table 1 molecules-25-03832-t001:** DPP IV IC_50_ of extracts and standards.

Extract or Standards	IC_50_ (µg/mL)
Cinnamon NADES	205.0
Sappan NADES	1254.0
Combined NADES Method-1	37.5
Combined NADES Method-2	353.6
Sappan Reflux	82.0
Brazilin	9.9
Sitagliptin	5 × 10^−3^

**Table 2 molecules-25-03832-t002:** The docking results of sitagliptin and marker compounds from cinnamon bark and sappan wood.

Ligand Compound	ΔG (kcal/mol)	Inhibition Constant	Interaction
Sitagliptin	−9.60	92.10 nM	Arg125, Glu205, Glu206, Val207, Ser209, Phe357, Arg358, Tyr547, Ser630, Tyr631, Val656, Trp659, Tyr662, Asp663, Tyr666, Asn710, Val711, His740
Brazilin	−6.35	22.06 µM	Glu205, Glu206, Ser209, Arg356, Phe357, Arg358, Tyr666, Arg669, Tyr670
Coumarin	−5.46	99.97 µM	Arg125, Tyr547, Ser630, Tyr631, Tyr662, Val656, Trp659, Tyr666, Asn710, Val711, His740
*trans*-Cinnamaldehyde	−4.95	237.03 µM	Arg125, Tyr547, Ser630, Tyr631, Val656, Trp659, Tyr662, Tyr666, Asn710, Val711, His740
*trans*-Cinnamic acid	−4.16	898.84 µM	Arg125, Glu205, Tyr547, Ser630, Tyr631, Val656, Trp659, Tyr662, Tyr666, Val711, His740

**Table 3 molecules-25-03832-t003:** NADES-based ultrasonic-assisted extraction (NADES-UAE) conditions for extraction of marker compounds from the sappan wood and cinnamon bark combination.

Optimum Extraction Conditions	Method and Extraction Time	Choline Chloride–Glycerol Ratio	Sample–NADES Ratio	Water Added
Method I (optimum conditions for brazilin)	UAE, for 50 min	2:1 *w*/*w*	1:2 *w*/*w*	47.57%
Method II (optimum conditions for *trans*-cinnamaldehyde and coumarin)	UAE, for 30 min	2:1 *w*/*w*	1:8 *w*/*w*	20.00%

**Table 4 molecules-25-03832-t004:** The elution gradient for the simultaneous analysis of the plant extract combination.

No	Time (min)	ddH_2_O (Acetic Acid)	Acetonitrile
1	0	84.5% (0.3%)	14.5%
2	8	74.5% (0.3%)	24.5%
3	16	50.0% (0.04%)	50.0%
4	24	40.0% (0.04%)	60.0%

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
