# Peer review of "Simultaneous Natural Deep Eutectic Solvent-Based Ultrasonic-Assisted Extraction of Bioactive Compounds of Cinnamon Bark and Sappan Wood as a Dipeptidyl Peptidase IV Inhibitor"

_molecules, 2020, doi:10.3390/molecules25173832_

Round 1
Reviewer 1 Report
Minor remarks
Avoid the use of third-person plural (for instance, we do that). Use the third-person singular for the scientific paper.
Use italic letters for “trans” and “in silico” in the manuscript.
Use SI units. For instance, use “min” for “minutes”.
The manufacturers’ cities should be provided for the used chemicals.
In section 4.5, the data about the used column are mentioned again. You previously described it in section 4.2. It is not necessary to duplicate these things.
Use the uniform term for HPLC (high-pressure liquid chromatography or high-performance liquid chromatography). You use the wrong term “high liquid performance chromatography”.
Also, you use the different terms for “DPP4” and “DPP IV”.
In Figure 2, are you sure that the ordinate is alright? Maybe should be introduced “absorbance (mAU)”? Also, the unit for the axis is not correct. Retention time (min).
In Figure 3, ppm is not visible for “trans-cinnamaldehyde”.
In Table 5, use the same decimal places.
In Table 6, use the same decimal places in both columns.
References have some errors that should be corrected. All corrections are given in the manuscript.
Major remarks
The advantages of the use of eutectic solvents is well described in the manuscript, but the significance of ultrasound-assisted extraction method should be more detailed discussed in the Introduction. It was reported in the papers DOI: 10.3390/antiox8080248; DOI: 10.1007/s13197-020-04312-w, so that they can be included in the Introduction.
In the manuscript, there are some sentences that should be improved because they are almost the same as in the other available literature. These sentences are clearly marked in the manuscript.
Lines 281-292...These sentences should be deleted because they do not belong to this manuscript.

Author Response
Response to Reviewer 1 Comments
Point 1: Minor remarks:
Avoid the use of third-person plural (for instance, we do that). Use the third-person singular for the scientific paper.
Use italic letters for “trans” and “in silico” in the manuscript.
Use SI units. For instance, use “min” for “minutes”.
The manufacturers’ cities should be provided for the used chemicals.
In section 4.5, the data about the used column are mentioned again. You previously described it in section 4.2. It is not necessary to duplicate these things.
Use the uniform term for HPLC (high-pressure liquid chromatography or high-performance liquid chromatography). You use the wrong term “high liquid performance chromatography”.
Also, you use the different terms for “DPP4” and “DPP IV”.
In Figure 2, are you sure that the ordinate is alright? Maybe should be introduced “absorbance (mAU)”? Also, the unit for the axis is not correct. Retention time (min).
In Figure 3, ppm is not visible for “trans-cinnamaldehyde”.
In Table 5, use the same decimal places.
In Table 6, use the same decimal places in both columns.
References have some errors that should be corrected. All corrections are given in the manuscript.
Response 1: we have been revised (with red highlight in the manuscript)
Point 2: Mayor remarks
The advantages of the use of eutectic solvents is well described in the manuscript, but the significance of ultrasound-assisted extraction method should be more detailed discussed in the Introduction. It was reported in the papers DOI: 10.3390/antiox8080248; DOI: 10.1007/s13197-020-04312-w, so that they can be included in the Introduction.
In the manuscript, there are some sentences that should be improved because they are almost the same as in the other available literature. These sentences are clearly marked in the manuscript.
Lines 281-292...These sentences should be deleted because they do not belong to this manuscript.
Response 2:
the significance of ultrasound-assisted extraction has been adden in lines 59-66 from some references include DOI: 10.3390/antiox8080248; DOI: 10.1007/s13197-020-04312-w
some sentence (lines 34-44) has been changed.
Lines 281-292 has been deleted.
Thank you for the valuable comments to improve the quality of our articles.

Reviewer 2 Report
The manuscript is devoted to the study of the antidiabetic activity of Cinnamon bark (Cinnamomum burmannii) and sappan wood (Caesalpinia sappan). The authors clearly show the advantage of natural deep eutectic solvent (a composition of choline chloride-glycerol) for the extraction of marker compounds. It has been shown that both individual extracts and their mixtures possess inhibitory activity against one of the main targets of diabetes therapy (DPP-4). But not all marker compounds have this activity. The article is well written and read with interest. Despite the overall good impression, there are a number of errors and typos that need to be corrected:
line 130: instead of 6.11 it is better to write 6 or 6.1 (redundant precision)
Line 170: in Figure 4, under the letter E, trans-cinnamate is indicated, and trans-cinnamaldehyde is re-drawn (and possibly calculated).
lines 281-292 this text is a complete quote from the rules for authors
In addition, I consider it desirable to note in the conclusion that it seems interesting to further study the extract from cinnamon bark to identify the active DPP-4 inhibiting component
Author Response
Point 1: The manuscript is devoted to the study of the antidiabetic activity of Cinnamon bark (Cinnamomum burmannii) and sappan wood (Caesalpinia sappan). The authors clearly show the advantage of natural deep eutectic solvent (a composition of choline chloride-glycerol) for the extraction of marker compounds. It has been shown that both individual extracts and their mixtures possess inhibitory activity against one of the main targets of diabetes therapy (DPP-4). But not all marker compounds have this activity. The article is well written and read with interest. Despite the overall good impression, there are a number of errors and typos that need to be corrected:
line 130: instead of 6.11 it is better to write 6 or 6.1 (redundant precision)
Response 1: we have been revised (lines 137 in revised manuscript
Line 170: in Figure 4, under the letter E, trans-cinnamate is indicated, and trans-cinnamaldehyde is re-drawn (and possibly calculated).
lines 281-292 this text is a complete quote from the rules for authors
Response 2: Lines 281-292 has been deleted.
In addition, I consider it desirable to note in the conclusion that it seems interesting to further study the extract from cinnamon bark to identify the active DPP-4 inhibiting component
Response 3: we have been added in conclusions (with red highlight in the manuscript)
Thank you for the valuable comments to improve the quality of our articles.
